# High PD-L1/IDO-2 and PD-L2/IDO-1 Co-Expression Levels Are Associated with Worse Overall Survival in Resected Non-Small Cell Lung Cancer Patients

**DOI:** 10.3390/genes12020273

**Published:** 2021-02-15

**Authors:** Vienna Ludovini, Fortunato Bianconi, Annamaria Siggillino, Jacopo Vannucci, Sara Baglivo, Valeria Berti, Francesca Romana Tofanetti, Maria Sole Reda, Guido Bellezza, Martina Mandarano, Maria Laura Belladonna, Giulio Metro, Rita Chiari, Angelo Sidoni, Francesco Puma, Vincenzo Minotti, Fausto Roila

**Affiliations:** 1Medical Oncology Division, Santa Maria della Misericordia Hospital, Piazzale Menghini 8/9, 06132 Perugia, Italy; annamaria.siggillino@ospedale.perugia.it (A.S.); sara.baglivo@ospedale.perugia.it (S.B.); francesca.tofanetti@ospedale.perugia.it (F.R.T.); mariasole.reda@ospedale.perugia.it (M.S.R.); giulio.metro@ospedale.perugia.it (G.M.); vincenzo.minotti@ospedal.perugia.it(V.M.); fausto.roila@ospedale.perugia.it (F.R.); 2Umbria Digitale, Regional Government of Umbria, Via G.B. Pontani 39, 06128 Perugia, Italy; fortunato.bianconi@gmail.com; 3Department of Thoracic Surgery, University of Perugia, Piazza Lucio Severi 1, 06132 Perugia, Italy; jacopo.vannucci@uniroma1.it (J.V.); wally.berti@gmail.com(V.B.); francesco.puma@unipg.it (F.P.); 4Section of Anatomic Pathology and Histology, Department of Medicine and Surgery, University of Perugia, Piazza Lucio Severi 1, 06132 Perugia, Italy; guido.bellezza@unipg.it (G.B.); mandaranomartina@gmail.com (M.M.); angelo.sidoni@unipg.it (A.S.); 5Department of Medicine and Surgery, University of Perugia, Piazza Lucio Severi 1, 06132 Perugia, Italy; marialaura.belladonna@unipg.it; 6Division of Medical Oncology, Ospedali Riuniti Padova Sud, via Albere 30, 35043 Monselice, Italy; rita.chiari@aulss6.veneto.it

**Keywords:** prognostic markers, early-stage NSCLC, immune-related genes, mRNA expression levels

## Abstract

Programmed death ligand 1 (PD-L1) expression is a predictive biomarker of the success of PD-1/PD-L1 inhibitor therapy for patients with advanced non-small cell lung cancer (NSCLC) but its role as a prognostic marker for early-stage resectable NSCLC remains unclear. We studied gene expression levels of immune-related genes PD-1, PD-L1, PD-L2, IDO-1, IDO-2 and INFγ in tumor tissue of surgically resected NSCLC and correlated the finding with clinicopathological features and patient outcomes. A total of 191 consecutive early-stage NSCLC patients who underwent curative pulmonary resection were studied. The mRNA expression levels of immune-related genes were evaluated by quantitative reverse transcription polymerase chain reaction (qRT-PCR) using RT^2^ Profiler PCR Arrays (Qiagen). PD-1, PD-L2 and IDO-2 gene expression levels were significantly higher in patients with squamous histology (*p* = 0.001, *p* = 0.021 and *p* < 0.001; respectively). PD-1, PD-L1 and IDO-2 gene expression levels were significantly higher in patients with higher stage (*p* = 0.005, *p* = 0.048 and *p* = 0.002, respectively). The univariate analysis for recurrence-free survival (RFS) and overall survival (OS) showed that patients with higher levels of three-genes (PD-L1/PD-L2/INFγ) (hazard ratio (HR)) 1.90 (95% confidence interval (CI), 1.13–3.21), *p* = 0.015) were associated with a worse RFS, while patients with higher levels of both genes (PD-L1/IDO-2) or (PD-L2/IDO-1) were associated with a worse OS (HR 1.63 95% CI, 1.06–2.51, *p* = 0.024; HR 1.54 95% CI, 1.02–2.33, *p* = 0.04; respectively). The multivariate interaction model adjusted for histology and stage confirmed that higher levels of three genes (PD-L1/PD-L2/INFγ) were significantly associated with worse RFS (HR 1.98, *p* = 0.031) and higher levels of both genes (PD-L1/IDO-2) and (PD-L2/IDO-1) with worse OS (HR 1.98, *p* = 0.042, HR 1.92, *p* = 0.022). PD-L1/IDO-2 and PD-L2/IDO-1 co-expression high levels are independent negative prognostic factors for survival in early NSCLC. These features may have important implications for future immune-checkpoint therapeutic approaches.

## 1. Introduction

Lung cancer is the leading cause of cancer-related mortality worldwide, with an estimated 1.3 million new cases each year [1,2]. Non-small-cell lung cancer (NSCLC) constitutes approximately 80% of all lung cancer cases and has a 5-year survival rate of only 15–20% [3]. Immune checkpoint inhibitors have begun to revolutionize the survival prospects of cancer patients [4,5,6], particularly those blocking the PD-1/PD-L1 (programmed cell death 1/programmed cell death 1 ligand 1) axis, which have yielded objective response rates of about 20% and are currently approved by the Food and Drug Administration (FDA) for a subset of patients with advanced disease [7,8,9,10]. Importantly, in patients with advanced NSCLC and PD-L1 expression on at least 50% of tumor cells, PD-1 inhibition was associated with longer progression-free and overall survival than platinum-based chemotherapy [11]. However, large cohorts of patients do not display a clinical response to PD-1/PD-L1 axis inhibition at late stages in disease progression. This has been attributed to several potential mechanisms, including low PD-L1 expression, T cell exclusion from tumor islets (cold tumors), and T cell dysfunction that emerges in the context of chronic antigen exposure [12,13,14,15,16]. Early-stage disease presents with a more intact immune system and a lower tumor burden, possibly affording immune checkpoint blockade the potential to confer a more favorable outcome. However, there are limited data on the prevalence and the prognostic role of PD-L1 expression in early-stage NSCLC. Data from small previously published studies is mixed with some showing poor prognosis and others with no prognostic significance [17,18,19,20]. PD-1 has two binding ligands, PD-L1 (B7-H1, CD274) and PD-L2 (B7-DC, CD273), both of which belong to the B7 family. PD-L2 is mainly expressed on activated dendritic cells and macrophages. PD-L1 is not only broadly expressed on non-immune cells, such as T cells, B cells, macrophages, and dendritic cells, but is also upregulated after their activation. Evaluation of PD-L1 positivity by conventional immunohistochemistry (IHC) is not well defined and subject to antibody and assay variability and interpretative subjectivity. In addition, the specificity and reproducibility of commercially available antibodies has not been thoroughly assessed. The expression and biological role of additional potentially actionable immune targets beyond PD-L1 in lung cancer are not well understood. A mechanism to escape the immunosurveillance could be played by indoleamine 2,3-dioxygenase 1 (IDO-1) that has been described in several human cancers. IDO-1 is a 42-45KD enzyme which catalyzes the rate limiting steps of tryptophan (Trp) degradation along the kynurenine pathway [21]. IDO-1 exerts a potent immunosuppressive effect through inhibiting T-lymphocytes and other immune cells; additionally, IDO-1 has been shown to induce favorable tumor progression in animal models of lung cancer [22,23]. Variable levels of IDO-1 have been found in human solid tumors including melanomas, gliomas, and carcinomas from different locations. Blockade of IDO-1 using small molecule inhibitors in combination with immune checkpoint blockade induces prominent antitumor responses in mouse models and reversal of tumor-associated immunosuppression by 1methyl-D-tryptophan appears to be dependent on host IDO-1 expression [24]. Furthermore, a cytokine such as interferon-γ (INFγ) is critical for innate and adaptive immunity. Once antigen-specific immunity develops, INFγ is secreted by activated effector T cells [25]. INFγ upregulates Major Histocompatibility Complex (MHC) class I and class II molecules and promotes antigen presentation on tumor cells [26]. With these functions, INFγ was expected to work as an antitumor agent. INFγ is also known to upregulate PD-L1 expression on tumor cells [26]. In mouse melanoma models, INFγ secreted from CD8-positive T cells was reported to upregulate PD-L1 [27]. In addition, INFγ can upregulate expression of other key immune suppressive molecules such as IDO-1 within the tumor microenvironment. Tumor adaptation takes advantage of this delicate balance of positive and negative immune signaling factors, allowing the cancer to survive and progress. Although the assessment of PD-L1 expression on tumor and immune cells can be useful to predict clinical response to PD-1 checkpoint blockade, it offers only limited insight into the biology of the tumor-immune interface. In particular, PD-L1 expression might represent only a component of T cell–related biology that is relevant to a favorable tumor immune microenvironment (TIM). Newer genomic technologies can be used to evaluate complexities of tumor and host immune cell interactions within the tumor microenvironment, going beyond the measurement of single analytes such as PD-L1. Zheng et al. [28] recently demonstrated a signature based on the B7-CD28 family that can predict lung adenocarcinoma patient prognosis. Nevertheless, their investigations were limited to B7-CD28 family members, which may not represent the status of the entire TIM. Therefore, it is essential to develop an immune signature on the basis of a comprehensive list of immune-related genes that can stand for the immune status of TIM and have prognostic ability in lung cancer. Our efforts concentrated on developing an immune signature with prognostic ability based on the comprehensive list of immune-related genes (PD-1, PD-L1, PD-L2, IDO-1, IDO-2 and INFγ) and then examining gene expression in the tumor microenvironment. To this end, we used RNA isolated from fresh tumor tissue samples of resected NSCLC patients and evaluated the correlation of gene expression levels with clinicopathological parameters and their impact on recurrence-free survival (RFS) and overall survival (OS).

## 2. Materials and Methods

### 2.1. Patient Selection

This study included consecutive patients with a histological diagnosis of stage I–III NSCLC who underwent pulmonary resection of the primary tumor in the period from 2009 to 2015 and who were then followed on a regular basis in a specific follow-up program. Human lung cancers and their corresponding non-tumor tissue (normal lung) were collected after surgery and were instantly put in a solution containing RNAlater (Qiagen S.p.A., Milan, Italy), then were frozen in liquid nitrogen and stored at −80 °C refrigeration. Samples had to contain at least 50% tumor cells to be eligible for array quantitative reverse transcription polymerase chain reaction (qRT-PCR) analysis as determined by one reference pathologist (G.B.) on adjacent separate sections. The pathological stage [29] was reassigned according to the 8th TNM staging and lung tumor histology was reclassified according to the 2015 World Health Organization (WHO) classification for lung tumors [30]. Follow-up including annual computed tomography (CT) scans of the chest and abdomen and chest X-rays in the 3 to 6 months intervals between the annual CT was planned for all patients in the first 3 years. None of the patients had prior anti-PD-1/PD-L1 therapies, neoadjuvant chemotherapy or EGFR/ALK-targeted therapy. Patients with stage II/III disease may have been offered adjuvant chemotherapy and patients with recurrent disease received chemotherapy and/or EGFR-targeted therapy. The study was approved by the local Ethics Committee (Number 2216/13 of Comitato Etico Aziende Sanitarie (CEAS) Umbria) and was conducted in accordance with ethical principles of the latest version of the Declaration of Helsinki. Written informed consent for gene expression analyses was obtained from each patient entering the study.

### 2.2. RNA Extraction and Array Quantitative Reverse Transcription Polymerase Chain Reaction (qRT-PCR)

Total RNA was extracted from frozen tumor tissues and normal lung after thawing and homogenizing by IKA Ultra-Turrax and QIAzol Lysis Reagent. RNA was extracted with the phenol chloroform method. From the aqueous phase, RNA was automatically purified by QiaCube instrument using miRNeasy Mini Kit according to the manufacturer’s instructions (Qiagen, Milan, Italy). RNA was eluted in 50 μL of RNase-free water and stored at −80 °C until use. The quality, integrity and quantity of the total RNA was evaluated on Experion™ Bioanalyzer (Biorad Technologies, Italy). Quantification of mRNA expression levels of PD-1, PD-L1, PD-L2, IDO-1, IDO-2 and INFγ were performed in triplicates by qRT-PCR using RT^2^ Profiler PCR Arrays (Qiagen, Milan, Italy). For each array, 1μg of RNA was reverse transcribed using Super Array’s first strand cDNA synthesis kit. Each real-time PCR reaction was performed with cDNA synthesized from 9 ng of RNA using Super Array’s RT2 Real-Time™ SYBR Green/ROX PCR Master Mix (PA-012) in a 25 μL reaction volume on the ABI 7300 Real-Time PCR System. Each PCR array also included stringent controls to monitor RNA quality by assessing reverse transcription efficiency, genomic DNA contamination to ensure the reliability of the PCR Array data and three housekeeping genes: glyceraldehyde 3-phosphate dehydrogenase (GAPDH), hypoxanthine phosphoribosyltransferase 1 (HPRT) e la β-actin, (ACTB). Nuclease-free water was used as template control. All reagents were dispensed automatically on the Hamilton Starlet platform. The gene expression results were calculated using the 2^−ΔΔCt^ method [31] considering as calibrator the normal lung tissues of each patient and as internal reference gene the housekeeping HPRT as it has shown a lower variability (mean Ct value of 26.7 (range 25.7–27.6), standard deviation (SD = 0.51) with respect to GAPDH (mean Ct value of 21.1 (range 18.8–22.1, SD = 0.82) and ACTB (mean Ct value of 18.8 (range 16.6–20.1, SD = 1.02).

### 2.3. Statistical Analysis

In order to assess the discriminatory accuracy of the mRNA gene expressions for predicting OS endpoint, we performed a bootstrap procedure to select the optimal cut-off point (COP). We used the regular Leaveone-out and 2000 repeats of the bootstrap [32]. The COP indicates which value of decreased or increased expression is relevant for the discrimination for overall survival. Based on these COPs, the two groups were divided into two subgroups which showed an expression rate over or under the COP. Continuous genes expression fold changes were correlated with the Kendall rank method and the significative test was adjusted using Bonferroni. Fisher’s exact test and Pearson х^2^ test were used to assess the association between immune-related gene expression groups based on COPs and clinical features. OS was calculated from the date of surgery to the date of death, and RFS was calculated from the date of surgery to the date of progression/recurrence or date of last follow-up. Survival analyses were performed according to the Kaplan–Meier method. Comparison of survival between groups was performed with the log-rank test. Cox’s proportional-hazard analysis was used for univariate and multivariate analysis to explore the effect of variables on survival. Statistical analyses were performed with R (3.6.2) was used for all statistical analyses, and a *p*-value of 0.05 was considered significant.

## 3. Results

### 3.1. Patient Characteristics 

From April 2008 to February 2015, 191 patients with radically resected NSCLC, referred to the Department of Thoracic Surgery, Perugia University Hospital, Italy, were recruited. Histological diagnosis was confirmed independently by three pathologists (G.B., M.M. and A.S.) at the Section of Anatomic Pathology and Histology, Department of Medicine and Surgery, University of Perugia, Italy. Patient characteristics are reported in Table 1. Median age at diagnosis was 67.6 years (range, 38.7–84.3), the majority of patients were male (n.137/191, 71.7%), former or current smokers (n.175/191, 91.6%), with good ECOG-PS (0: n.188/191, 98.4%), poorly differentiated histology (n.81/191, 42.4%) and stage I disease (n.101/191, 52.9%). Histological types included 120 (62.8%) adenocarcinoma (ADC), 69 (36.1%) squamous cell carcinoma (SCC), and 2 (1.1%) adenosquamous carcinoma. The following resections were carried out: 183 lobectomies (95.8%), 4 pneumonectomies (2.1%) and four wedge resections (2.1%) with hilar and mediastinal lymph node dissection. Following surgery, 49 patients (25.6%) were treated with chemotherapy, 12 (6.3%) with radiotherapy, 4 (3.1%) with chemoradiotherapy and the remaining 130 (68.1%) received no adjuvant treatment. Eighty-six (45%) patients relapsed after surgery, 105 (55%) had no relapse. Ninety-six (50.3%) patients died between 1 and 133 months from initial diagnosis; among them 62 (64.6%) died from lung carcinoma.

### 3.2. PD-1, PD-L1, PD-L2, IDO-1, IDO-2 and INFγ Gene Expression and Association with Clinic-Pathological Characteristics

PD-1, PD-L1, PD-L2, IDO-1, IDO-2 and INFγ gene expression was evaluated in 191 patients. The mean value of PD-1 gene expression was 3.72 (range 0.16 to 68.55), of PL-L1 was 7.82 (range 0.24 to 88.10), of PD-L2 was 6.28 (range 0.65 to 59.22), of IDO-1 was 6.25 (range 0.11 to 149.20), of IDO-2 was 6.11 (range 0.06 to 92.21), of INFγ was 3.04 (range 0.10 to 66.70). When adopting the optimal overall cut-off point (COP) that was four times the value of calibrator *Fold-change*, 40 patients (20.9%) had higher levels of PD-1, 114 (59.7%) had higher levels of PD-L1, 88 (46.3%) had higher levels of PD-L2, 81 (42.4%) had higher levels of IDO-1, 61 (31.9%) had higher levels of IDO-2 and 34 (17.8%) had higher levels of INFγ. The frequency of expression levels of all immune-related genes and their associations with the clinic-pathological variables of patients are shown in Table 2. PD-1, PD-L1 and IDO-2 gene expression levels were significantly higher in patients with higher TNM stage (*p* = 0.005, *p* = 0.048 and *p* = 0.002, respectively). PD-1, PD-L2 and IDO-2 gene expression levels were significantly higher in those with squamous histology (*p* = 0.001, *p* = 0.021 and *p* < 0.001; respectively). PD-L2 gene expression was significantly higher in patients aged <60 years (*p* = 0.016). We also assessed the correlations among the expression levels of immune-related genes. As a continuous variable all the immune-related genes PD-1, PD-L1, PD-L2, IDO-1, IDO-2 and INFγ were significantly correlated with each other with *p* value < 0.001 (Figure 1). The same using the categorized variable according to COPs, there was a significant association between PD-L1 high expression and IDO-1 and IDO-2 (x^2^ = 38.0 *p* < 0.001, x^2^ = 17.3 *p* < 0.001, respectively), as well as between PD-L2 high expression and IDO-2 and IDO-1 (x^2^ = 24.5 *p* < 0.001, x^2^ = 45.4 *p* < 0.001, respectively). Both PD-L1 and PD-L2 high expression were significantly associated with INF**γ** (x^2^ = 17.04 *p* < 0.001; x^2^ = 38.4 *p* < 0.001, respectively). The same association was found between IDO-1, IDO-2 high expression and INF**γ** (x^2^ = 35.5 *p* < 0.0001; x^2^ = 8.4 *p* < 0.0001, respectively).

### 3.3. Prognostic Value of PD-1, PD-L1, PD-L2, IDO-1, IDO-2 and INFγ Gene Expression 

Survival data in this study were taken on 30 November 2019. The median follow-up time was 66 months (m) (range: 1.67 to 133 m) while 60 (34.5%) patients had cancer related deaths. Eighteen (18.9%) of the 95 patients still on follow-up experienced recurrence: local recurrence was observed in 7 patients (7.37%), lung recurrence and other sites in 11 patients (11.6%). No statistically significant difference in RFS was observed among patients with higher or lower levels for PD1 (*p* = 0.52) or PD-L1 (*p* = 0.35), or PD-L2 (*p* = 0.11), or IDO-1 (*p* = 0.27), or IDO-2 (*p* = 0.20), or INFγ (*p* = 0.08). Interestingly, the patients with higher levels of three-genes (PD-L1/PD-L2/INFγ) showed a shorter RFS than patients with lower levels (hazard ratio (HR) 1.90 (95% confidence interval (CI),1.13–3.21), *p* = 0.01 (Table 3, Figure 2). There was no relevant difference in OS among patients with higher or lower levels for PD1 (*p* = 0.29) or PD-L1 (*p* = 0.08), or PD-L2 (*p* = 0.12), or IDO-1 (*p* = 0.17), or IDO-2 (*p* = 0.06), or INFγ (*p* = 0.26); whereas patients with higher levels of both genes (PD-L1/IDO-2) or (PD-L2/IDO-1) showed a shorter OS than patients with lower levels (HR 1.63 95%CI, 1.06–2.51, *p* = 0.02; HR 1.54 95%CI, 1.02–2.33, *p* = 0.04; respectively) (Table 4, Figure 3). At univariate analysis, stage III and adenocarcinoma histology were significantly associated with worse RFS (*p* = 0.01 and *p* = 0.03, respectively), whereas male and stage II-III were significantly associated with worse OS (*p* = 0.04 and *p* = 0.02, respectively). A multivariate Cox regression model for OS and RFS was built using the variables that were found significant at univariate analysis. Higher levels of both genes (PD-L1/IDO-2) (HR 1.98; 95% CI 1.02–3.83, *p* = 0.04), and both genes (PD-L2/IDO-1) (HR 1.92; 95% CI 1.10–3.35, *p* = 0.02), adjusting for stage (HR 1.65; 95% CI 1.03–2.63, *p* = 0.03) were significantly associated with worse OS (Table 4). Higher levels of three genes (PD-L1/PD-L2/INFγ) (HR 1.98; 95% CI 1.06–3.71, *p* = 0.03), adjusting for histology (HR 2.16; 95% CI 1.30–3.6, *p* = 0.003) and stage (HR 2.13; 95% CI 1.2–3.8, *p* = 0.01) were significantly associated with worse RFS. 

We have also performed the univariate and multivariate survival analyses for RFS and OS of immune-related genes expression with respect to histological types.

In the adenocarcinomas group, the univariate analysis showed that the patients with higher levels of PD-L2 (*p* = 0.02), IDO-2 (*p* = 0.005) and INFγ (*p* = 0.01) had a shorter RFS than patients with lower levels. Interestingly, the patients with higher levels of two-genes (PD-L1/IDO-2; *p* < 0.001), (PD-L2/IDO-1; *p* = 0.04) and three-genes (PD-L1/PD-L2/INFγ; *p* = 0.005) showed a shorter RFS than patients with lower levels (Appendix A). Similarly, the univariate analysis for OS showed that the patients with higher levels of PD-L1 (*p* = 0.04), IDO-1 (*p* = 0.05), two-genes (PD-L1/IDO-2; *p* = 0.02), (PD-L2/IDO-1; *p* = 0.03) and three-genes (PD-L1/PD-L2/INFγ; *p* = 0.04) had a worst OS than patients with lower levels (Appendix A). However, these statistically significant differences were not confirmed in the multivariate analysis.

In the subgroup of patients with squamous carcinoma, the univariate analysis showed no statistically significant difference in RFS and OS among patients with higher or lower levels of immune-related genes expression. 

## 4. Discussion

This study was carried out to explore the relationship between gene expression levels of immune-related genes (i.e., PD-1, PD-L1, PD-L2, IDO-1, IDO-2 and INFγ) assessed by quantitative-PCR arrays and their association with prognosis on a series of resected NSCLC patients. Only a few reports have evaluated gene expression of immune-related genes in lung cancer and no data exist in resected NSCLC [33]. In the last few years, PD-L1 expression in lung cancer has been mainly studied at the protein level using IHC as in our previous study [34]; however, divergent results have been reported, notably regarding its prognostic value [35]. Such divergence has often been related to the usual limitations of IHC such as the absence of standardization for PD-L1, mainly in terms of specificity and reproducibility of available antibodies, the definition of positivity cutoff, and interpretative subjectivity. Our study at the mRNA level allowed us to avoid these limitations and we also evaluated a fairly large series of NSCLC patients. To the best of our knowledge, with 191 patients analyzed who had a longer follow-up (>5 years), this is the first comprehensive analysis of the expression levels of immune-related genes in early NSCLC to evaluate their association with survival. From a biological point of view, our results show that PD-1, PD-L2 and IDO-2 gene expression levels were significantly higher in patients with squamous cell carcinoma and PD-1, PD-L1 and IDO-2 gene expression levels were significantly higher in patients with higher TNM stage. A positive association between high levels of PD-L1 or PD-L2 and IDO-1 or IDO-2 was also observed. From a clinical point of view, the subset of patients with the concomitant higher levels of genes (PD-L1/PD-L2/INFγ) showed a relative risk of 1.98 for reduced RFS, suggesting that relapse to disease was more associated with higher levels of concomitant genes rather than with expression levels of individual genes. Our results also showed that the concomitant higher levels of both PD-L1/IDO-2 and PD-L2/IDO-1 gene expression were independent negative prognostic factors for OS in resected NSCLC patients. As shown in Figure 4, IDO signaling pathways exert an immune suppressive function by the activation of T-reg cells and the anergy of T effector cells. PD-L1/PD-1 inhibits T effector cell proliferation and stimulate T-reg cell proliferation while INFγ upregolates PD-L1, PD-L2 and IDO expression. An immune suppressive micro-environment promotes tumor growth, an increasing risk of progression (poor RFS) and an unfavorable prognosis (worse OS). Taken together, it is conceivable that patients with higher levels of both PD-L1/IDO-2 and PD-L2/IDO-1 gene expression would benefit more from immune checkpoint inhibitor therapy, while the key to a higher response rate is to restore preexisting immunity. Based on this finding, a dual combination of NSCLC therapy, such as inhibitors of both PD-1/PD-L1 immune checkpoint and IDO-2, or IDO-1, might be hypothesized. Currently, the combination of immune checkpoint inhibitors and IDO-1 hinders has already been tested in ongoing clinical trials, with encouraging results in NSCLC patients [36,37]. Among the factors successfully used in immunotherapy are PD-1, PD-L1 and PD-L2 which belong to the immunological checkpoint system. In several studies these targets have shown to inhibit the functions of T cell receptor (TCR) and their functional activity due to the PD-1/PD-L1/PD-L2 [38]. Moreover, from the literature data, it emerges that the silencing of PD-L1 accelerates the antitumor immune responses and enhances the anti-cancer capacity of dendritic cells [39]. Under neoplastic conditions, the expression on PD-L1 tumor cells strongly correlates with the increased risk of progression and with an unfavorable prognosis as reported in our study. Various components of the immune system have been shown to be determining factors during cancer initiation and progression. Evading immune destruction has been recognized as an emerging hallmark of cancer. A meta-analysis by Ma G, et al. [40] conducted on a total of 25 articles with 5861 patients showed that the expression of PD-L1 is a prognostic factor related to poor survival in NSCLC. A study by Zhou C, et al. [41], evaluating PD-L1 on 108 radically resected NSCLC patients, showed the expression of PD-L1 was an independent prognostic factor for reduced survival, particularly in those with non-squamous histotype. The study of Wu S. et al. [42] evaluated the expression of PD-L1 by IHC and at mRNA level by in situ hybridization, and showed that overexpression of PD-L1 is more common in male patients and smokers with lung adenocarcinoma and that PD-L1 expression was a poorer prognostic factor in patients with surgically resected lung adenocarcinoma. Also in our study, we observed that only in the subgroup of patients with adenocarcinoma, higher levels of immune-gene expression alone or in combination had a redution effect of RFS and OS.

Regarding the cytokine INFγ, some studies show the determining role of this factor for the expression of PD-L1 on tumor cells. Also, intra-tumoral infiltration of T cells can improve the probability of response to anti-PD-1 therapies as reported in the study by Ayers, M [43]. IFN-γ is confirmed to possess immune-activating properties and hence is proven to have anti-cancer effects in vivo. In addition, some studies have also indicated that IFN-γ has the ability to directly inhibit cancer cell growth. Inflammation-induced PD-L1 expression by INFγ differs from oncogene-induced PD-L1 expression in that PD-L1 expression depends on the time and site of the immune response. Our result showed that patients with concomitant higher levels of both genes (PD-L1/PD-L2/INFγ) showed poor RFS. Therefore, we could speculate that in cases with poor outcomes, including those with three genes PD-L1/PD-L2/INFγ expression, inflammation-induced PD-L1 expression may be dominant, and an immunosuppressive state in relation to over-production of neutrophils may have occurred. At the same time, PD-1-mediated tumor immune escape by which cancer cells can become progressive may also have been activated. Preclinical and clinical studies have indicated that the production of INFγ by CD8+ T cells induces PD-L1 expression on tumor-resident cells [44,45] which is consistent with our findings. Interesting considerations regarding other molecules involved in immunosuppression caused by the tumor emerged from the literature data. Among these, IDO-1 was evaluated as a further target for immunotherapeutic intervention by the evidence reported in some studies showing its pro-tumorigenic effect [46]. Regarding the expression of IDO-1 and IDO-2, several studies show that their role can influence tumor progression [47]. In particular, the study by Lindström, V, et al. showed that increased IDO-1 activity in patients with chronic lymphatic leukemia may affect disease progression [48]. Liu Y, et al. [49] conducted an IDO-2 study to evaluate the impact of gene silencing as a way to inhibit B16-BL6 tumor cells in a mouse model. The authors show that the silencing of IDO-2, achieved through a small interfering RNA (siRNA), inhibits the proliferation of tumor cells, stops the cell cycle in G1, induces greater cell apoptosis and reduces cell migration in vitro. IDO represents a potent important immunoregulatory enzyme capable of creating a suppressive microenviroment in human tumor through several mechanisms which act synergistically to promote tumor growth and survival of cancer cells. The concomitant higher levels of both PD-L1/IDO-2 and PD-L2/IDO-1 gene expression associated to worse OS emerged from our results is consistent with the suppressive immunoregulatory role of IDO signalling. Limitations of our study include: first, its retrospective nature cannot avoid the potential confounding biases despite the large sample size. Second, similar to other studies [50,51], gene expression signatures are subject to sampling bias caused by intratumor genetic heterogeneity. Third, patients in this cohort were not treated with immune checkpoint inhibitors, thus the predictive value of the signature for immunotherapy could not be directly evaluated. Further research will be carried out aiming to gain further knowledge of the unknown molecular mechanisms involved in immune-escape and the development of more standardized and reproducible molecular expression evaluation methods that exceed the limits of the availability of fresh tissue, as in our case, and the known instability of RNA.

## 5. Conclusions

In conclusion, this study generates an immune-relate gene signature that can not only predict NSCLC patient survival outcome but also reflects the immune status of lung cancer. It is probable that it is not a single parameter or a single marker that determines the prognosis of NSCLC, as for other neoplasms, but it more likely depends on a dynamic balance between immunosuppression and antitumor response. This signature can be clinically used for the improvement of patient OS, individualized therapy methods based on the risk score and possible response to immunotherapy. Prospective studies are needed to further validate its analytical accuracy for estimating prognoses and to test its clinical utility in individualized management of NSCLC. 

## Figures and Tables

**Figure 1 genes-12-00273-f001:**
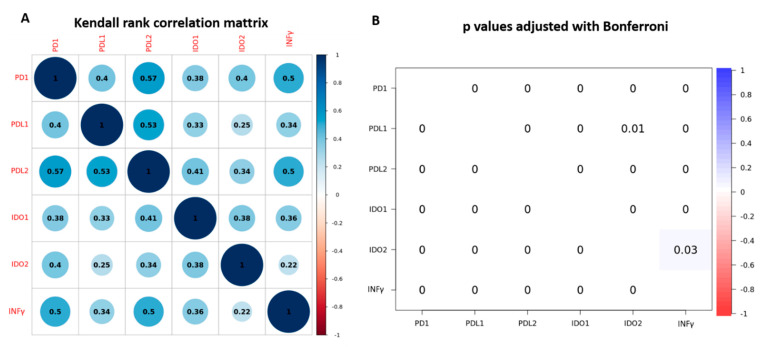
Correlation among the expression levels of immune-related genes. (**A**). Kendal rank correlation matrix. (**B**). Probability significative test was adjusted using Bonferroni.

**Figure 2 genes-12-00273-f002:**
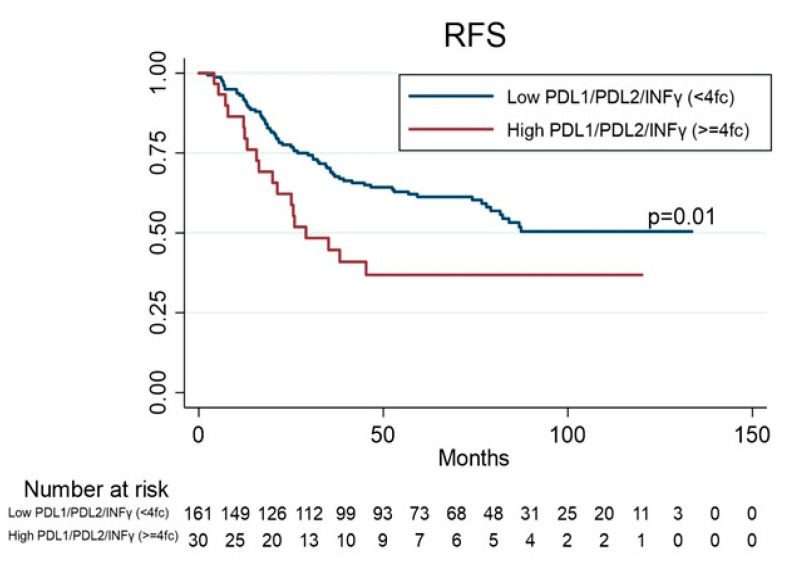
Kaplan–Meier estimates for recurrence-free survival (RFS) according to PD-L1/PD-L2/INFγ co-expression levels.

**Figure 3 genes-12-00273-f003:**
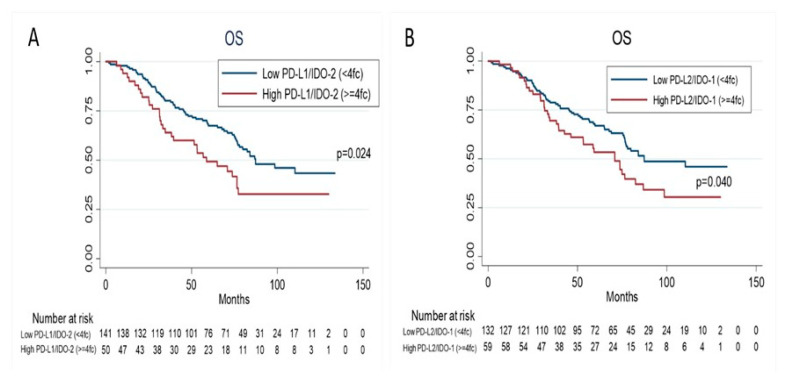
Kaplan–Meier estimates for overall survival (OS) according to PD-L1/IDO-2 co-expression levels (**A**), and PDL-2/IDO-1 co-expression levels (**B**).

**Figure 4 genes-12-00273-f004:**
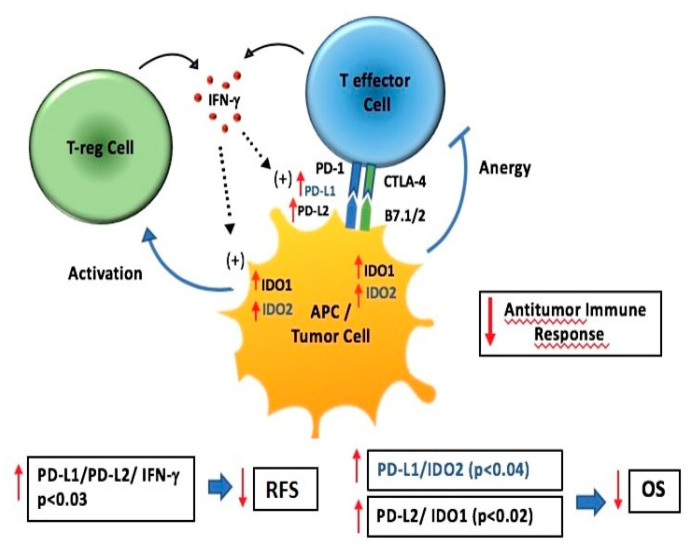
IDO-1, IDO-2, PD-L1, PD-L2 and INFγ interactions within the tumor microenviroment. IDO signaling pathways exert immune suppressive function by the activation of T-reg cell and the anergy of T effector cell. PD-L1/PD-1 inhibits T effector cell proliferation and stimulates T-reg cell proliferation while INFγ upregolates PD-L1, PD-L2 and IDO expression. Immune suppressive microenviroment promotes tumor growth increasing risk of progression (poor RFS) and unfavorable prognosis (worse OS).

**Table 1 genes-12-00273-t001:** Patient characteristics.

Characteristics	Patients	
	*N* = 191	%
**Median Age**, years (range)	67.6 (38.7–84.3)	
**Gender**		
Female	54	28.3
Male	137	71.7
**Smoking History**		
Never smokers	16	8.4
Former smokers	76	39.8
Current smokers	99	51.8
**Histology**		
Invasive Adenocarcinomas	120	62.8
Squamous cell carcinomas	69	36.1
Adenosquamous carcinomas	2	1.1
**Grading**		
1	11	5.8
2	99	51.8
3	81	42.4
**pStage**		
I	101	52.9
II	56	29.3
III	34	17.8
**Type of Resection**		
Lobectomy	183	95.8
Pneumonectomy	4	2.1
Other	4	2.1
**Relapse**		
No	105	55.0
Yes	86	45.0
**Exitus**		
Live	95	49.7
Dead	96	50.3

**Table 2 genes-12-00273-t002:** Clinicopathological variables of patients according to PD-1, PD-L1, PD-L2, IDO-1, IDO-2, INFγ gene expression levels.

Variables	PD-1, N. (%)		PD-L1, N. (%)		PD-L2, N. (%)		IDO-1, N. (%)		IDO-2, N. (%)		INFγ, N. (%)	
	Low	High	Low	High	Low	High	Low	High	Low	High	Low	High
**N**	151 (79.1)	40 (20.9)	77 (40.3)	114 (59.7)	103 (53.4)	88 (46.1)	110 (57.6)	81 (42.4)	130 (68.1)	61 (31.9)	157 (82.2)	34 (17.8)
**Age (years)**												
<60	24 (12.6)	10 (5.2)	11 (5.8)	23 (12.0)	12 (6.3)	22 (11.5)	16 (8.4)	18 (9.4)	22 (11.5)	12 (6.3)	25 (13.1)	9 (4.7)
≥60	127 (66.5)	30 (15.7)	66 (34.6)	91 (47.6)	91 (47.6)	66 (34.6)	94 (49.2)	63 (33.0)	108 (56.5)	49 (25.7)	132 (69.1)	25 (13.1)
	*p* = 0.181		*p* = 0.297		*p* = 0.016		*p* = 0.170		*p* = 0.643		*p* = 0.145	
**Sex**												
Female	45 (23.6)	9 (4.7)	18 (9.4)	36 (18.9)	32 (16.8)	22 (11.5)	31 (16.2)	23 (12.0)	42 (22.0)	12 (6.3)	46 (24.1)	8 (4.2)
Male	106 (55.5)	31 (16.2)	59 (30.9)	78 (40.8)	71 (37.2)	66 (34.5)	79 (41.4)	58 (30.4)	88 (46.1)	49 (25.6)	111 (58.1)	26 (13.6)
	*p* = 0.362		*p* = 0.217		*p* = 0.353		*p* = 0.974		*p* = 0.071		*p* = 0.675	
**Smoking**												
Never	15 (7.9)	1 (0.5)	6 (3.1)	10 (5.2)	8 (4.2)	8 (4.2)	7 (3.7)	9 (4.7)	14 (7.3)	2 (1.1)	14 (7.3)	2 (1.1)
Current	59 (30.9)	17 (8.9)	30 (15.7)	46 (24.1)	40 (20.9)	36 (18.9)	44 (23.0)	31 (16.8)	48 (25.1)	28 (14.7)	62 (32.5)	14 (7.3)
Former	77 (40.3)	22 (11.5)	41 (21.5)	58 (30.4)	55 (28.8)	44 (23.0)	59 (30.9)	40 (20.9)	68 (35.6)	31 (16.2)	81 (42.4)	18 (9.4)
	*p* = 0.388		*p* = 0.975		*p* = 0.882		*p* = 0.502		*p* = 0.162		*p* = 0.920	
**Histology**												
Invasive Adenocarcinomas	104 (54.4)	16 (8.4)	54 (28.3)	66 (34.6)	73 (38.2)	47 (24.6)	73 (38.2)	47 (24.6)	95 (49.7)	25 (13.1)	100 (52.4)	20 (10.5)
Squamous	45 (23.6)	24 (12.6)	23 (12.0)	46 (24.1)	29 (15.2)	40 (21.0)	36 (18.9)	33 (17.3)	33 (17.3)	36 (18.9)	55 (28.8)	14 (7.3)
Adenosquamous	2 (1.0)	0 (0.0)	0 (0.0)	2 (1.0)	1 (0.5)	1 (0.5)	1 (0.5)	1 (0.5)	2 (1.0)	0 (0.0)	2 (1.0)	0 (0.0)
	***p* = 0.001**		*p* = 0.146		***p* = 0.021**		*p* = 0.498		***p* < 0.001**		*p* = 0.702	
**Grading**												
1	8 (4.2)	3 (1.6)	2 (1.0)	9 (4.7)	5 (2.6)	6 (3.2)	3 (1.6)	8 (4.2)	7 (3.7)	4 (2.1)	7 (3.7)	4 (2.1)
2	81 (42.4)	18 (9.4)	43 (22.5)	56 (29.3)	55 (28.8)	44 (23.0)	61 (31.9)	38 (19.9)	74 (38.7)	25 (13.1)	82 (43)	17 (8.9)
3	62 (32.5)	19 (9.9)	32 (16.8)	49 (25.7)	43 (22.5)	38 (19.9)	46 (24.1)	35 (18.3)	49 (25.7)	32 (16.7)	68 (35.6)	13 (6.8)
	*p* = 0.540		*p* = 0.299		*p* = 0.818		*p* = 0.094		*p* = 0.107		*p* = 0.261	
**p-Stage**												
I	89 (46.6)	12 (6.3)	49 (25.7)	52 (27.2)	59 (30.9)	42 (22.0)	65 (34.0)	36 (18.9)	80 (41.9)	21 (11.0)	87 (45,5)	14 (7.3)
II	39 (20.4)	17 (8.9)	18 (9.4)	38 (19.9)	28 (14.7)	28 (14.7)	31 (16.2)	25 (13.1)	31 (16.2)	25 (13.1)	45 (23.6)	11 (5.8)
III	23 (12.0)	11 (5.8)	10 (5.2)	24 (12.6)	16 (8.3)	18 (9.4)	14 (7.3)	20 (10.5)	19 (9.9)	15 (7.9)	25 (13.1)	9 (4.7)
	***p* = 0.005**		***p* = 0.048**		*p* = 0.404		*p* = 0.059		***p* = 0.002**		*p* = 0.208	

Abbreviations: PD-1, programmed cell death 1; PD-L1, programmed cell death 1 ligand 1; PD-L2, programmed cell death 1 ligand 2; IDO-1, indoleamine 2,3-dioxygenase 1; IDO-2, indoleamine 2,3-dioxygenase 2; INFγ, Interferon-γ; significant *p* value ≤ 0.05 in bold.

**Table 3 genes-12-00273-t003:** Univariate and multivariate analyses for recurrence-free survival (RFS).

			RFS					
			Univariate Analysis			Multivariate Analysis		
**Variables**		N. Pts at risck	HR	95% CI	* *p*	HR	95% CI	* *p*
**Age**	<60 (Ref)	34						
	≥60	157	0.84	0.49–1.44	0.54			
**Sex**								
	F (Ref)	54						
	M	137	1.03	0.65–1.65	0.88			
**Smoking**								
	Never (Ref)	16						
	Current	76	2.09	0.82–5.30	0.12			
	Former	99	1.58	0.62–4.00	0.33			
**Histology**								
	Squamous (Ref)	69						
	Adenocarcinoma	120	1.62	1.02–2.58	**0.03**	2.16	1.30–3.60	**0.003**
	Adenosquamous	2	1.39	0.18–10.25	0.32	1.36	0.17–10.46	0.76
**p-Stage**								
	I (Ref)	101						
	II	56	1.71	1.05–2.79	**0.03**	2.13	1.26–3.62	**0.005**
	III	34	2.01	1.17–3.41	**0.01**	2.13	1.20–3.80	**0.01**
**PD-1**								
	Low < 4fc (Ref)	151						
	High ≥ 4fc	40	1.17	0.70–21.96	0.52			
**PD-L1**								
	Low < 4fc (Ref)	77						
	High ≥ 4fc	114	1.22	0.79–1.90	0.35			
**PD-L2**								
	Low < 4fc (Ref)	103						
	High ≥ 4fc	88	1.41	0.92–2.15	0.11			
**IDO-1**								
	Low < 4fc (Ref)	110						
	High ≥ 4fc	81	1.26	0.832–1.93	0.27			
**IDO-2**								
	Low < 4fc (Ref)	130						
	High ≥ 4fc	61	1.33	0.85–2.08	0.20			
**INFγ**								
	Low < 4fc (Ref)	157						
	High ≥ 4fc	34	1.55	0.93–2.59	0.08			
**PDL1/PDL2/INFγ**								
	Low < 4fc (Ref)	161						
	High ≥ 4fc	30	1.90	1.13–3.21	**0.01**	1.98	1.06–3.71	**0.03**

Abbreviations: RFS, Recurrence-free Survival; HR, Hazard Ratio; CI, Confidence Interval; PD-1, programmed cell death 1; PD-L1, programmed cell death 1 ligand 1; PD-L2, programmed cell death 1 ligand 2; IDO-1, indoleamine 2,3-dioxygenase 1; IDO-2, indoleamine 2,3-dioxygenase 2; IFNγ, Interferon-γ; * *p* value ≤ 0.05 in bold.

**Table 4 genes-12-00273-t004:** Univariate and multivariate analyses for overall survival (OS) (Cox’s proportional hazard regression model).

			OS					
			Univariate Analysis			Multivariate Analysis		
Variables		N. Pts at risk	HR	95% CI	* *p*	HR	95% CI	* *p*
**Age**								
	<60	34						
	≥60	157	1.04	0.60–178	0.88			
**Sex**								
	F (Ref)	54						
	M	137	1.62	1.00–1.64	**0.04**	1.53	0.94	0.08
**Smoking**								
	Never (Ref)	16	1					
	Current	76	1.81	0.77–4.28	0.17			
	Former	99	1.62	0.69–3.78	0.26			
**Histology**								
	Squamous (Ref)	69						
	Adenocarcinoma	120	1.14	0.75–173	0.53			
	Adenosquamous	2	0.93	0.12–6.85	0.94			
**p-Stage**								
	I (Ref)	101	1					
	II	56	1.68	1.07–2.64	**0.02**	1.65	1.03–2.63	**0.03**
	III	34	1.55	0.90–2.66	0.10	1.46	0.83–2.57	0.18
**PD-1**								
	Low < 4fc (Ref)	151						
	High ≥ 4fc	40	1.26	0.78–2.04	0.29			
**PD-L1**								
	Low < 4fc (Ref)	77						
	High ≥ 4fc	114	1.45	0.95–2.21	0.08			
**PD-L2**								
	Low < 4fc (Ref)	103						
	High ≥ 4fc	88	1.37	0.91–2.04	0.12			
**IDO-1**								
	Low < 4fc (Ref)	110						
	High ≥ 4fc	81	1.31	0.88–1.96	0.17			
**IDO-2**								
	Low < 4fc (Ref)	130						
	High ≥ 4fc	61	1.47	0.97–2.22	0.06			
**INFγ**								
	Low < 4fc (Ref)	157						
	High ≥ 4fc	34	1.33	0.80–2.20	0.26			
**PD-L1/IDO-2**								
	Low < 4fc (Ref)	141						
	High ≥ 4fc	50	1.63	1.06–2.51	**0.02**	1.98	1.02–3.83	**0.04**
**PD-L2/IDO-1**								
	Low < 4fc (Ref)	132						
	High ≥ 4fc	59	1.54	1.02–2.33	**0.04**	1.92	1.10–3.35	**0.02**

Abbreviations: OS, Overall Survival; HR, Hazard Ratio; CI, Confidence Interval; PD-1, programmed cell death 1; PD-L1, programmed cell death 1 ligand 1; PD-L2, programmed cell death 1 ligand 2; IDO-1, indoleamine 2,3-dioxygenase 1; IDO-2, indoleamine 2,3-dioxygenase 2; IFNγ, Interferon-γ; * *p* value ≤ 0.05 in bold.

## Data Availability

Not applicable.

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
