# Peer review of "High PD-L1/IDO-2 and PD-L2/IDO-1 Co-Expression Levels Are Associated with Worse Overall Survival in Resected Non-Small Cell Lung Cancer Patients"

_genes, 2021, doi:10.3390/genes12020273_

Round 1
Reviewer 1 Report
Both the design and methodology of the study were chosen correctly and prove the presented thesis. The prepared figures are legible and consistent with the description. The text is comprehensible and free of major grammatical or stylistic errors.
Author Response
Response to Reviewer 1
We thank the reviewer for his comments as he did not ask for any changes to our manuscript.

Reviewer 2 Report
Dr. Ludovini and co-authors have reported that PD-L1/IDO-2 and PD-L2/IDO-1 co-expression levels in tumors in NSCLC correlate with postoperative prognosis. This manuscript is well-written and may have very important information that can contribute to improved outcomes with immune checkpoint inhibitors (ICIs). Although ICIs have not been performed, we believe that this information will help in the appropriate selection of patients whose survival may be improved by ICIs.
In order to make this manuscript more fruitful, some remarks and comments are noted below.
- the authors use the term "DFS." We assume that most authors usually use recurrence-free survival (RFS) when describing postoperative survival status. If accepted, please replace this term with RFS.
- 120 adenocarcinomas were enrolled in this cohort. Did the authors include adenocarcinoma in situ or minimally invasive adenocarcinoma? These adenocarcinomas, especially AIS, probably have little or no immunological response to the immune system, either locally or systemically. It may be desirable to exclude these GGN-predominant adenocarcinomas.
- Table 2 should be supplemented with pathologic information, if possible, indicating the malignant and invasive nature of the tumor cells, such as lymphovascular or pleural invasion, and compared with the expression of each immunogenic gene. Similarly, T and N factors should also be involved.
- page 10 mentions "Higher levels of both genes (PD-L1/IDO-2) (HR1.98; 95% CI 1.02–3.83, p=0.04). Both genes (PD-L2/IDO-1) (HR 1.92; 95% CI 1.10–3.35, p=0.02), adjusting for stage (HR 1.65; 95% CI 1.03–2.63, p=0.03) were significantly associated with worse OS (Table 4)." This statement should be revised.
- the number of patients in each of Tables 3 and 4 and in the figures should be provided. In particular, the "number of patients at risk" in the survival curves is very important for understanding the results.
- my interest is the difference in immunogenicity between adenocarcinoma and squamous cell carcinoma. As mentioned by the authors, several gene expressions were significantly different between adenocarcinomas and squamous cell carcinomas. It would be very interesting to see how these gene expressions affect the prognosis in each histological type. Univariate and multivariate (single and combined) survival analyses of immunogenic gene expression in respective histological type should be performed and reported.
Author Response
Response to Editorial Office
Please find attached a letter explaining the modifications of the manuscript according to the comments of the referee.
The text has been changed and further linguistic revision has been performed so the changes have been highlighted in yellow.
Response to Reviewer 2 Comments
Point 1:the authors use the term "DFS." We assume that most authors usually use recurrence-free survival (RFS) when describing postoperative survival status. If accepted, please replace this term with RFS.
Response 1: We agree with review and we changed the word DFS with RFS in all article and also in Fig. 2, Fig. 4 and Table 3
Point 2:120 adenocarcinomas were enrolled in this cohort. Did the authors include adenocarcinoma in situ or minimally invasive adenocarcinoma? These adenocarcinomas, especially AIS, probably have little or no immunological response to the immune system, either locally or systemically. It may be desirable to exclude these GGN-predominant adenocarcinomas.
Response 2: We thank the referee for the observation, but all adenocarcinomas enrolled in our study were invasive adenocarcinoma. We added in Table 1 the word invasive adenocarcinoma.
Point 3: Table 2 should be supplemented with pathologic information, if possible, indicating the malignant and invasive nature of the tumor cells, such as lymphovascular or pleural invasion, and compared with the expression of each immunogenic gene. Similarly, T and N factors should also be involved.
Response 3: We thank the reviewer for the advice regarding the table 2. The biological behavior and prognosis of the neoplasms studied — we analyzed only invasive squamous cell carcinomas, adenocarcinomas or adenosquamous carcinomas — relies upon the p-Stage of the disease, which is already reported in Table 2. Moreover, the T and N factors, together with the pleural invasion, contribute to the definition of p-Stage itself. For the abovementioned reason, it is redundant to correlates these parameters with the expression of each immunogen gene. Finally, the histological evaluation of lymphovascular invasion, although recommended for clinical care, is not a parameter required yet in the staging of NSCLCs and for the patients’ risk stratification, accordingly to the current official guidelines (Rami-Porta R, et al. 2017. Lung. Pag. 431-456 In: Amin MB, Edge SB, Greene FL, et al. AJCC cancer Staging Manual. 8th ed. New York, NY: Springer 2017New York, NY: Springer 2017).
Point 4: page 10 mentions "Higher levels of both genes (PD-L1/IDO-2) (HR1.98; 95% CI 1.02–3.83, p=0.04). Both genes (PD-L2/IDO-1) (HR 1.92; 95% CI 1.10–3.35, p=0.02), adjusting for stage (HR 1.65; 95% CI 1.03–2.63, p=0.03) were significantly associated with worse OS (Table 4)." This statement should be revised.
Response 4: We agree with the referee and we change the sentence
Point 5: the number of patients in each of Tables 3 and 4 and in the figures should be provided. In particular, the "number of patients at risk" in the survival curves is very important for understanding the results.
Response 5: We agree with the referee and we added the number of patients at risk in Table 3 and 4 and in survival curves.
Point 6: my interest is the difference in immunogenicity between adenocarcinoma and squamous cell carcinoma. As mentioned by the authors, several gene expressions were significantly different between adenocarcinomas and squamous cell carcinomas. It would be very interesting to see how these gene expressions affect the prognosis in each histological type. Univariate and multivariate (single and combined) survival analyses of immunogenic gene expression in respective histological types should be performed and reported.
Response 6: We agree with the referee and we performed the Univariate and Multivariate survival analyses of immune-related genes expression in respective histological types and the results are reported in Table S1 and Table S2 for adenocarcinomas histology, whereas for the patients with squamous cell carcinomas, we did not show the data since the univariate analysis proved no statistically significant difference in RFS and OS concerning patients with higher or lower levels of immune-related genes expression. Moreover, in paragraph 3.3 “Prognostic value of PD-1, PD-L1, PD-L2, IDO-1, IDO-2 and INFγ gene expression” we added the sentence:
We have also performed the univariate and multivariate survival analyses for RFS and OS of immune-related genes expression with respect to histological types.
In the adenocarcinomas group, the univariate analysis showed that the patients with higher levels of PD-L2 (p=0.02), IDO-2 (p=0.005) and INFγ (p=0.01) had a shorter RFS than patients with lower levels. Interestingly, the patients with higher levels of two-genes (PD-L1/IDO-2; p<0.001), (PD-L2/IDO-1; p=0.04) and three-genes (PD-L1/PD-L2/INFγ; p=0.005) showed a shorter RFS than patients with lower levels (Table S1). Similarly, the univariate analysis for OS showed that the patients with higher levels of PD-L1 (p=0.04), IDO-1 (p=0.05), two-genes (PD-L1/IDO-2; p=0.02), (PD-L2/IDO-1; p=0.03) and three-genes (PD-L1/PD-L2/INFγ; p=0.04) had a worst OS than patients with lower levels (Table S2). However, these statistically significant differences were not confirmed in the multivariate analysis.
In the subgroup of patients with squamous cell carcinomas, the univariate analysis showed no statistically significant difference in RFS and OS among patients with higher or lower levels of immune-related genes expression (Data not shown).
Moreover in paragraph 4. Discussion, we added the sentence:” Also in our study, we observed that only in the subgroup of patients with adenocarcinoma, higher levels of immune-gene expression alone or in combination did they have an effect of reduced RFS and OS.
